# Instance-Level Costs for Nuanced Classifier Evaluation

**Kabir Kang** [1]  **Stephen Mussmann** [1]

## Abstract

Standard classification treats all errors equally, but in applications such as content moderation and medical screening, mistakes on clear-cut cases are more costly than errors on ambiguous ones. From a contextual bandit framework, we propose normalized excess cost (NEC), a metric that weighs classification errors by per-example costs and reduces to standard error rate when costs are uniform. Costs can derive from annotator vote margins, distance from decision thresholds, or confidence ratings. Across text, image, and tabular benchmarks, we find that NEC is often substantially lower than error rate—models with 5% error rate can achieve 1.8% NEC—revealing that most mistakes concentrate on ambiguous, low-cost examples. We also find that incorporating costs into training via loss weighting, sampling strategies, or regression yields inconsistent benefits. Our framework provides a practical methodology for deriving and evaluating instance-level misclassification costs, even if cost-sensitive training offers limited benefit.

## 1. Introduction

Standard accuracy treats all mistakes as equally undesirable, but in practice this is often untrue. In content moderation and medical screening, misclassifying a clear-cut case is more serious than erring on a borderline example. This gap between uniform error counting and real-world consequences motivates evaluation and training methods that weight errors by their importance.

We propose a framework that assigns each example a misclassification cost reflecting its importance. This framing connects classification evaluation to decision theory: the cost-weighted error we minimize corresponds exactly to

regret in a contextual bandit. These costs can be generated by annotator votes, distance to a threshold, or directly by annotator ratings.

While existing work studies instance-dependent misclassification costs (Höppner et al., 2022; Bahnsen et al., 2015; Vanderschueren et al., 2022), in all cases, it's only used where there are objective, quantifiable costs, such as in fraud detection or direct marketing. Unlike these works, we highlight the importance of modeling costs even when they are not as clearly definable. We show that the introduced *normalized excess cost* (NEC) is often significantly lower than the naive error, indicating that most errors are on borderline (low-cost) examples.

While we establish the construction of and evaluation with instance-level costs as providing a different, more nuanced view of errors, we also present a negative result for using these costs during training, as is done in other work (Bahnsen et al., 2015; Zadrozny et al., 2003). In particular, we try strategies such as upsampling or upweighting points based on costs, only training on the highest cost points, or directly regressing the (signed) costs for classification, but find the improvement is mixed or non-existent. This raises the paradox that instance-level costs significantly change evaluation but are unable to change training, at least through straightforward approaches.

**Contributions.**

1. A general framework for deriving and using per-example misclassification costs in the normalized excess cost (NEC), equivalent to the normalized regret.

2. Experiments on image, text, tabular, and synthetic datasets showing NEC is significantly lower than standard misclassification error (equivalent to NEC with uniform costs).

3. An experimental comparison of training methods that leverage instance-level costs.

## 2. Related Work

### 2.1. Cost-Sensitive Classification

Cost-sensitive learning addresses classification problems where different types of errors incur different penalties.

[1] School of Computer Science, Georgia Institute of Technology, Atlanta, GA, USA. Correspondence to: Kabir Kang <kkang68@gatech.edu>.

*Proceedings of the $43^{rd}$ International Conference on Machine Learning*, Seoul, South Korea. PMLR 306, 2026. Copyright 2026 by the author(s).

Elkan (2001) established theoretical foundations showing that optimal decision thresholds can be derived from class probabilities and misclassification costs, and that resampling training data is equivalent to adjusting these thresholds. Zadrozny et al. (2003) extended this framework to example-dependent costs through cost-proportionate weighting, where training examples are weighted by their associated misclassification costs. Langford & Beygelzimer (2005) provided a reduction from cost-sensitive multiclass classification to binary classification with formal regret guarantees, showing that instance-dependent cost vectors can be accommodated within their framework.

These methods assume costs are *externally specified*—typically by domain experts or derived from objective quantities. Bahnsen et al. (2015) exemplify this approach in fraud detection, where the cost of missing a fraudulent transaction (a false negative) equals the transaction amount, while a false positive incurs a fixed administrative cost of investigating the flagged transaction. Such financially-grounded costs are instance-dependent but objectively measurable. Our work addresses a complementary setting: when costs reflect *subjective* importance such as those derived from human disagreement, where the cost signal must be extracted from annotation patterns rather than external metadata. This shifts the problem from "how to train given costs" to "how to derive and validate costs from disagreement."

A separate line of work questions whether instance weighting reliably changes what deep models learn. Byrd & Lipton (2019) examined importance weighting in deep learning, finding that its effect on over-parameterized models diminishes over successive epochs when the training data is separable. While their analysis does not consider disagreement-derived costs, it offers a prior explanation for why instance-level cost-weighting may fail to improve a converged model: on separable data, the weighting washes out. Our experiments complement theirs by examining cost-weighting derived from annotation margins, finding that its effectiveness depends critically on whether the cost signal is predictable from input features.

### 2.2. Human Label Variation and Task Ambiguity

A separate literature has examined settings where annotator disagreement reflects genuine ambiguity rather than noise. Dawid & Skene (1979) introduced the foundational EM-based approach for jointly estimating true labels and annotator reliability, treating disagreement as a nuisance to be marginalized. Plank et al. (2014) challenged this view, demonstrating that inter-annotator disagreement in part-of-speech tagging is systematic and linguistically motivated rather than erroneous. This perspective—that disagreement carries signal—has gained substantial traction.

Pavlick & Kwiatkowski (2019) showed that disagreements in natural language inference persist even with additional context, reflecting inherent ambiguity in the task itself. Peterson et al. (2019) collected full label distributions for CIFAR-10 and demonstrated that training on soft labels improves robustness and out-of-distribution generalization. Nie et al. (2020) scaled this approach with ChaosNLI, collecting 100 annotations per example and revealing that models achieve near-perfect accuracy on high-agreement examples but perform at chance on low-agreement ones. Uma et al. (2021) surveyed methods for learning from disagreement, cataloging approaches ranging from soft-label training to multi-annotator modeling. Plank (2022) synthesized these developments in a position paper arguing that human label variation should be embraced throughout the ML pipeline.

Most recently, Kurniawan et al. (2025) proposed soft evaluation metrics based on fuzzy set theory, where soft accuracy measures overlap between predicted and human judgment distributions. Their extensive empirical study found that two simple strategies for preserving annotator disagreement during training—using the full label distribution from the votes (*soft labels*), or treating each annotator's judgment as a separate training instance (*disaggregated annotations*)—outperform more complex approaches. However, soft metrics evaluate *calibration*—whether the model's output distribution matches the human distribution—rather than *decision quality*. Our metric instead treats disagreement magnitude as misclassification cost: a model that correctly classifies high-consensus examples while erring on ambiguous ones incurs lower expected cost than the reverse, even if both models have identical accuracy.

### 2.3. Predicting Disagreement

Several methods leverage predicted disagreement for purposes other than evaluation. Raghu et al. (2019) train models to predict which medical cases will elicit expert disagreement, using these predictions to route difficult cases for second opinions. Their key finding—that directly predicting uncertainty outperforms deriving it from classifier confidence—demonstrates that disagreement is feature-predictable. We build on this insight but pursue a different goal: rather than routing examples, we use disagreement magnitude to weight evaluation, asking whether models that minimize cost-weighted error differ from those minimizing unweighted error (i.e., standard misclassification error).

### 2.4. This Work's Position

Our work bridges cost-sensitive classification and ambiguity in classification, including human label variation. From the former, we adopt the framework of instance-dependent misclassification costs and expected-cost minimization. From

the latter, we take the insight that examples have varying levels of ambiguity and that annotator disagreement reflects meaningful signal about example ambiguity. The gap we address is that cost-sensitive methods assume costs are given, while disagreement methods use variation for training or calibration rather than cost-based evaluation. We propose deriving per-example costs when they aren't easily quantifiable, and evaluating models by normalized excess cost, connecting to decision-theoretic notions of regret.

## 3. Cost-Sensitive Classification

### 3.1. Motivation

Standard accuracy assumes every mistake is equally costly, yet for many practical tasks, some errors matter more than others. The challenge is that "importance" is often subjective: unlike fraud amounts or medical expenditures, there is no external ledger recording the cost of misclassifying a borderline content moderation case. We argue that even *approximate* costs are more informative than the implicit assumption of uniform costs. If annotators split 6–4 on whether a comment is toxic, either classification is defensible; if they agree 10–0, misclassification is a clear failure. Normalized Excess Cost (NEC) operationalizes this intuition without requiring that costs be precisely calibrated.

### 3.2. Decision-Theoretic Regret

We consider a binary action setting (actions $+1$ and $-1$) where each example consists of a triple $(x, r^{(-1)}, r^{(+1)})$. $x \in \mathcal{X}$ is the input, $r^{(-1)} \in \mathbb{R}$ is the reward if the $-1$ action is chosen and $r^{(+1)} \in \mathbb{R}$ is the reward if the $+1$ action is chosen. Our goal is to find a policy $\pi : \mathcal{X} \to \{-1, +1\}$ which minimizes the regret $R(\pi) = \mathbb{E}[\max_{a \in \{-1,+1\}} r^{(a)}] - \mathbb{E}[r^{(\pi(x))}]$.

Specific to binary actions, we can add additional notation to simplify the problem. For a given example, define $\Delta = r^{(+)} - r^{(-)}$. Then the regret is simply $R(\pi) = \mathbb{E}_{x,\Delta}[|\Delta| \cdot \mathbb{1}[\text{sign}(\Delta) \neq \pi(x)]]$. In this form, we see a close similarity to binary classification with a label $y = \text{sign}(\Delta)$ and an instance-level misclassification cost of $|\Delta|$. The metric of standard accuracy has $\Delta \in \{-1, 1\}$ and the metric of cost-sensitive classification (with false positive and false negative costs $a$ and $b$, respectively) has $\Delta \in \{-a, b\}$.

### 3.3. Classification Evaluation

In order to produce an interpretable metric with the same scale as the misclassification error, we scale the regret by the maximum possible regret. For a classifier's predictions $\hat{y}_i$, we refer to this quantity as the normalized excess cost

(NEC) where "excess cost" is synonymous with "regret",

$$\text{NEC} = \frac{\sum_{i=1}^{n} |\Delta_i| \cdot \mathbb{1}[\hat{y}_i \neq \text{sign}(\Delta_i)]}{\sum_{i=1}^{n} |\Delta_i|}. \qquad (1)$$

When $|\Delta_i| = 1$ for all $i$, NEC reduces to error rate, making the two metrics directly comparable.

$$\text{Error Rate} = \frac{1}{n} \sum_{i=1}^{n} \mathbb{1}[\hat{y}_i \neq \text{sign}(\Delta_i)]. \qquad (2)$$

A method which randomly guesses receives 0.5 NEC and 0.5 error rate on average.

The ratio of the Error Rate to the NEC characterizes how much better a model performs on high-cost examples relative to low-cost ones. When this ratio exceeds 1, the model's errors concentrate on ambiguous cases, which is a desirable property that error rate alone cannot detect. NEC measures the fraction of total cost incurred by misclassification. Lower NEC indicates better alignment with task-specific priorities: a model can achieve low NEC by correctly classifying high-cost examples even if it errs on many ambiguous, low-cost cases.

Our formulation provides a decision-theoretic grounding for instance-level classification costs. From a decision-theoretic perspective, high $|\Delta|$ indicates decisions with large reward gaps where correct classification matters greatly, while low $|\Delta|$ indicates near-indifference where either action yields similar reward. This interpretation is agnostic to cost provenance and holds whether costs derive from annotator disagreement, clinical thresholds, or any other source.

### 3.4. Cost Derivation

Our framework applies whenever per-example costs are available. We describe three natural sources of costs.

**Multi-annotator labels.** When multiple annotators label each example, their agreement provides a natural cost signal. Let $n_{i,\text{yes}}$ and $n_{i,\text{no}}$ denote the number of positive and negative votes for example $i$. The Laplace-smoothed estimate of the yes frequency is $(n_{i,\text{yes}} + 1)/(n_{i,\text{yes}} + n_{i,\text{no}} + 2)$. We define the reward difference as the log-odds of the smoothed yes vote estimate:

$$\Delta_i = \log \frac{n_{i,\text{yes}} + 1}{n_{i,\text{no}} + 1}, \qquad (3)$$

Note that Laplace smoothing ensures finite values when annotators unanimously agree. Log-odds is the natural scale for binary outcomes: it is symmetric around zero, unbounded, and $|\Delta|$ is monotonically related to distance from maximum uncertainty. High $|\Delta|$ indicates strong annotator consensus where misclassification is costly; $|\Delta| \approx 0$ indicates ambiguity where either label is defensible.

**Threshold-based labels.** When binary labels derive from thresholding a continuous measurement, the distance to the threshold provides a natural cost. Let $z_i$ be the continuous value and $\tau$ the decision threshold. We define:

$$\Delta_i = z_i - \tau, \qquad (4)$$

A patient whose measurement is far above or below the threshold represents a clear-cut case where misclassification is serious; a patient near the threshold is a borderline case where the label itself is less certain.

**Annotated rating.** When multiple annotators are not available, a single annotator can directly annotate the level of ambiguity and/or confidence. For example, ratings on a 5-point or 7-point scale can be easily converted to $\Delta$ values by subtracting the midpoint value.

### 3.5. Training with Costs

In addition to standard cross-entropy training, we compare several approaches that use the costs $|\Delta_i|$. Intuitively, cost-weighting encourages the model to prioritize high-cost examples during training. Since our evaluation metric weights errors by cost, aligning the training objective with the evaluation metric should improve performance—a principle analogous to importance weighting in domain adaptation (Shimodaira, 2000).

**Standard classification.** The standard approach minimizes binary cross-entropy loss on the label $\text{sign}(\Delta_i)$. Let $\hat{p}_i \in [0, 1]$ denote the model's predicted probability of the positive class $+1$, and then at inference, we predict $\hat{y}_i = +1$ if $\hat{p}_i \geq 0.5$, and $\hat{y}_i = -1$ otherwise. The standard training loss is,

$$\mathcal{L}_{\text{cls}} = -\sum_{i=1}^{n} \big[ \mathbf{1}[\Delta_i \geq 0] \log \hat{p}_i + \mathbf{1}[\Delta_i < 0] \log(1 - \hat{p}_i) \big]. \qquad (5)$$

**Cost-weighted classification.** To emphasize high-cost examples during training, we weight each example's loss by its cost:

$$\mathcal{L}_{\text{wt}} = -\sum_{i=1}^{n} |\Delta_i| \cdot \big[ \mathbf{1}[\Delta_i \geq 0] \log \hat{p}_i + \mathbf{1}[\Delta_i < 0] \log(1 - \hat{p}_i) \big]. \qquad (6)$$

Cost-weighting modifies only the training objective; at inference, we use the same decision rule of predicting $+1$ if and only if $\hat{p}_i \geq 0.5$.

**Sampling strategies.** Rather than modifying loss weights, we can alter the training distribution. We evaluate two approaches applied per-class to preserve class balance: (1)

*probability-proportional upsampling* (P_up), which resamples with replacement where selection probability is proportional to $|\Delta_i|$, maintaining the original dataset size while increasing the effective influence of high-cost examples; and (2) *top-k% filtering* (Tdown$k$), which retains only examples with $|\Delta_i|$ at or above the $(100-k)$th percentile within each class. We test $k \in \{30, 50, 70\}$, corresponding to keeping the highest-cost 30%, 50%, or 70% of examples respectively.

**$\Delta$-regression.** Rather than predicting a binary label, we train a regression model to predict the signed margin $\Delta_i$ directly using squared error loss:

$$\mathcal{L}_{\text{reg}} = \sum_{i=1}^{n} \big(\hat{\Delta}(x_i) - \Delta_i\big)^2. \qquad (7)$$

At inference, we classify by thresholding at zero: $\hat{y} = \text{sign}(\hat{\Delta}(x))$. Regression yields a continuous $\hat{\Delta}$ that can be used for ranking or downstream selective prediction, analogous to a calibrated confidence estimate.

## 4. Experiments

### 4.1. Experimental Setting

**Datasets.** We evaluate on four datasets spanning text, image, and tabular domains (Table 1).

*Jigsaw* contains online comments from the Civil Comments archive, annotated for toxicity by multiple crowd annotators (Borkan et al., 2019) and we use the Laplace-smoothed annotation log-odds as $\Delta_i$, as described in Section 3.4.

*Turkey* contains images of turkeys from barn surveillance cameras, labeled by multiple annotators for visible injuries (Volkmann et al., 2021; Schmarje et al., 2022), and we use the Laplace-smoothed annotation log-odds as $\Delta_i$.

*NHANES* contains tabular health data from the CDC's National Health and Nutrition Examination Survey (2013–2014) (National Center for Health Statistics (NCHS), 2014). The task is hypertension classification using the 2017 ACC/AHA guideline threshold of 130 mmHg systolic blood pressure, with $\Delta_i = \text{SBP}_i - 130$. Features include age, gender, race/ethnicity, and BMI.

*iNaturalist* contains plant images sampled from the iNaturalist API (iNaturalist, 2024), labeled by Gemini 2.5-pro (Google DeepMind, 2025) on a 7-point confidence scale from "indisputably wild" (1) to "indisputably cultivated" (7). We use $\Delta_i = 4 - \text{rating}_i$, demonstrating cost derivation from a single annotator's confidence rather than multi-annotator disagreement.

To understand the performance of methods in an idealized setting, we also construct a *Synthetic* dataset where the

signed margin is a linear function of the input features plus Gaussian noise. We use a unit weight vector $w \in \mathbb{R}^d$ (with $d = 50$) and isotropic Gaussian features $x_i \sim \mathcal{N}(0, I_d)$, and set

$$\Delta_i = w^\top x_i + \varepsilon_i, \qquad \varepsilon_i \sim \mathcal{N}(0, 0.1^2), \qquad (8)$$

Since the signal term $w^\top x_i$ has unit variance while the noise standard deviation is only 0.1, the cost $|\Delta_i|$ is almost entirely determined by the features—an idealized case in which costs are linearly predictable and cost-sensitive training should help.

| Dataset | $N$ | Input | Target | Cost source |
|---|---|---|---|---|
| Jigsaw | 1.8M | Text | Toxic | Human votes |
| Turkey | 8,040 | Image | Injured | Human votes |
| iNaturalist | 9,956 | Image | Wild | LLM rating |
| NHANES | 7,455 | Tabular | Hypertensive | Blood Pressure |

*Table 1.* Datasets used in experiments. Costs derive from multi-annotator disagreement or distance to a decision threshold.

**Models.** We use standard architectures appropriate to each domain: TF-IDF features and frozen RoBERTa (Liu et al., 2019) embeddings with logistic regression for Jigsaw, frozen ImageNet-pretrained ResNet-50 (He et al., 2016) embeddings with logistic regression for Turkey and iNaturalist, and HistGradientBoosting for NHANES tabular features. We also evaluate end-to-end fine-tuning of RoBERTa for Jigsaw and ImageNet-pretrained ResNet-50 for Turkey and iNaturalist.

**Implementation details.** All linear models use L2 regularization with default hyperparameters. We use an 80/10/10 train/validation/test split stratified on the label $\text{sign}(\Delta)$, and report results over 10 random seeds with mean $\pm 1.96$ times the standard error (corresponding to a 95% Gaussian confidence interval).

### 4.2. NEC vs. Error Rate

We first compare NEC against error rate across datasets and models, then examine how the two metrics scale with training-set size, how NEC relates to other evaluation metrics, and how robust it is to the choice of $\Delta$ transform. Section 4.3 then evaluates training methods that use the costs.

**NEC vs error rate.** Figure 1 and Table 2 present our main results comparing NEC and error rate across all datasets. The key finding is that *error rate and NEC can diverge substantially*, revealing different pictures of model performance.

On Jigsaw with TF-IDF, the model achieves 5.34% error rate but only 1.76% NEC—error rate is more than three

*Table 2.* Main results: NEC and Error Rate (%) for Standard cross-entropy training across datasets. Models with "-LP" use frozen pretrained embeddings with a linear classifier; "-FT" indicates end-to-end fine-tuning. Mean $\pm$ 95% CI over 10 seeds. Lower is better.

| Task | Model | NEC | Error |
|---|---|---|---|
| Jigsaw | TF-IDF | 1.8±0.0 | 5.3±0.0 |
| Jigsaw | RoBERTa-LP | 3.3±0.2 | 7.6±0.4 |
| Jigsaw | RoBERTa-FT | 1.8±0.0 | 4.7±0.0 |
| Turkey | ResNet50-LP | 4.2±0.6 | 6.7±0.6 |
| Turkey | ResNet50-FT | 2.3±0.2 | 4.8±0.2 |
| NHANES | HistGBM | 14.8±0.9 | 21.7±0.7 |
| iNaturalist | ResNet50-LP | 10.4±0.6 | 11.7±0.6 |
| iNaturalist | ResNet50-FT | 9.3±0.4 | 11.1±0.5 |
| Synthetic | LogReg | 0.5±0.0 | 4.6±0.2 |

times higher. This indicates that most errors occur on low-cost examples where annotators disagreed, while the model performs well on high-cost examples with strong annotator consensus.

The gap between error rate and NEC varies by dataset. On Turkey, error rate (6.68%) is $1.6\times$ NEC (4.16%). On NHANES, error rate (21.68%) is $1.5\times$ NEC (14.83%). On iNaturalist, the ratio is smaller: error rate (11.73%) is only $1.1\times$ NEC (10.39%). The Synthetic control shows the largest gap by construction (4.55% error rate vs. 0.50% NEC, a $9\times$ ratio), reflecting that its costs are linearly predictable from features and most errors land on low-cost examples near the decision boundary. Under fine-tuning, both metrics improve substantially—on Jigsaw, RoBERTa-FT reduces NEC from 3.29% to 1.78% (46% relative) and error rate from 7.56% to 4.73%; on Turkey, ResNet-FT reduces NEC from 4.16% to 2.27% and error rate from 6.68% to 4.75%—while the same per-dataset ordering is preserved (error-to-NEC ratio is $2.5\times$ on Jigsaw RoBERTa-FT, $2.1\times$ on Turkey, $1.2\times$ on iNaturalist), confirming that the size of the gap reflects the cost distribution of the task rather than the choice of frozen vs. end-to-end training. The $\Delta$ distributions for each dataset are shown in Figure 2.

From a practitioner's standpoint, the gap means a system reported at $5\%$ error rate can have substantially better cost-weighted performance than accuracy suggests—on Jigsaw, a $5\%$-error model is also a $1.8\%$-NEC model, since most errors land on the low-cost ambiguous cases that humans disagree on. The size of the gap also informs the task itself: a small gap (as on iNaturalist) indicates the model performs similarly on high cost and low cost examples, while a large gap (as on Jigsaw) signals many ambiguous examples where the model errs.

**Sample size scaling.** Figure 3 shows the error rate and NEC as a function of training set size on Jigsaw, varying $N$ from 1,000 to 100,000. Both metrics improve with more

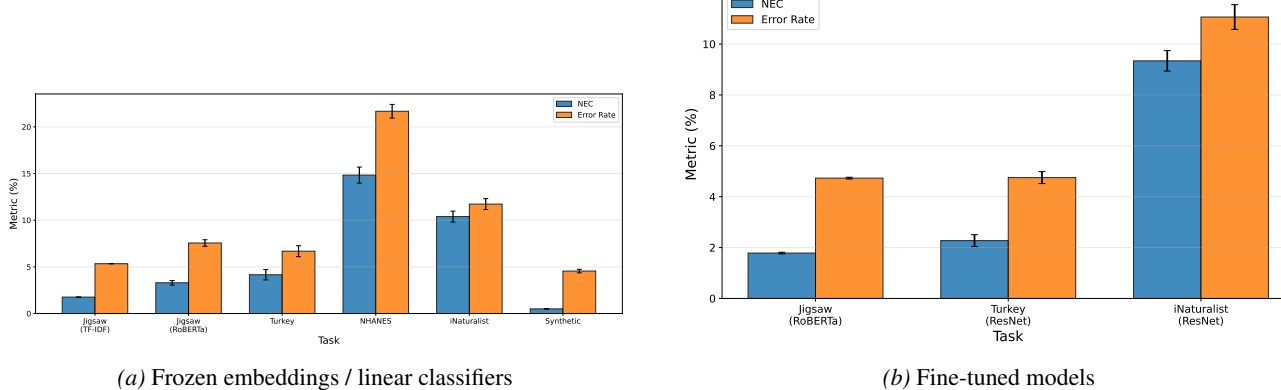

*(a)* Frozen embeddings / linear classifiers          *(b)* Fine-tuned models

*Figure 1.* NEC vs Error Rate across datasets. Blue bars show NEC (cost-weighted error); orange bars show standard error rate. Fine-tuning improves both metrics while preserving the NEC/error-rate gap.

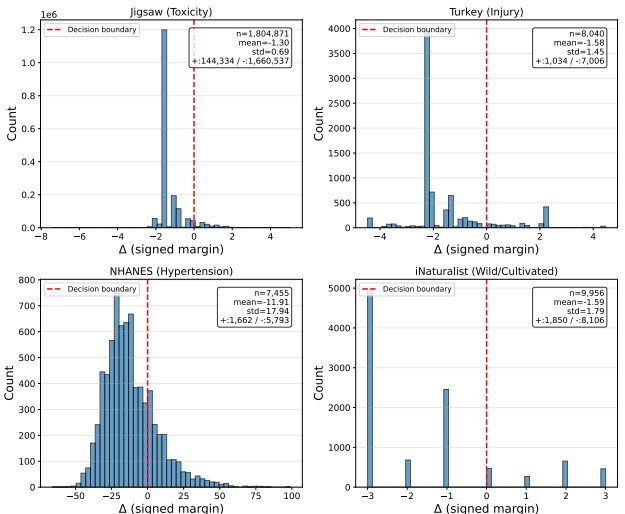

*Figure 2.* Distribution of signed $\Delta$ across datasets. All histograms are oriented so the minority class is on the right (positive) side. The red dashed line marks the decision boundary at $\Delta = 0$.

data, and the two move largely together.

The same approximate co-movement holds across model architectures, random seeds, and sampling strategies, suggesting that the ratio is largely a property of the task and its cost distribution rather than of the particular model.

**Comparison to soft and class-aware metrics.** Brier score (Brier, 1950) and log-loss measure prediction calibration in some form, which are "soft metrics" similar to NEC, though conceptually they are very different (e.g., NEC isn't a function of the predicted probabilities, only the binary predicted labels). Across our datasets, NEC has Spearman rank correlation with Brier of around $0.90$ and with log-loss of around $0.85$–$0.89$. Correlation with raw error rate is higher still ($\sim 0.95$–$0.98$), confirming that the NEC–error gap reported throughout this paper reflects *relative weighting* of

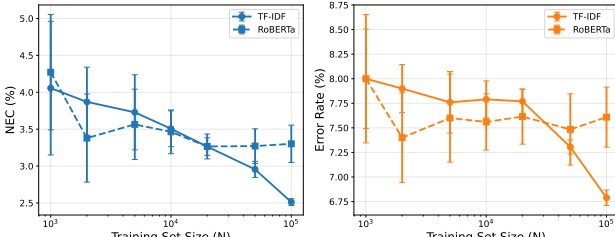

*Figure 3.* NEC and Error Rate as a function of training set size on Jigsaw. Both metrics decrease with more data and move largely together.

the same errors, not a different ranking of methods.

Where NEC diverges sharply is from class-aware metrics. On Jigsaw, the Spearman rank correlation between NEC and $1 - \text{BalAcc}$ (Brodersen et al., 2010) is $0.41$, and between NEC and $1 - F_2$ (van Rijsbergen, 1979) is $0.02$—essentially uncorrelated. The reason is that BalAcc and F-beta scores weight errors by class membership (positive vs. negative) rather than by per-example cost. In our datasets, the minority class tends to contain lower-$|\Delta|$ examples on average, so a method that does well on BalAcc by correctly classifying minority cases need not do well on NEC if those cases happen to be the ambiguous ones. Practitioners optimizing for class-balanced metrics in cost-sensitive settings risk a misleading signal: a model can be near-optimal under $F_2$ while incurring substantially more cost-weighted error than a baseline. When the goal is to minimize regret on high-cost cases, NEC and class-aware metrics are not interchangeable.

**Sensitivity to the $\Delta$ transform.** The absolute value of NEC depends on the choice of $\Delta$ transform. For multi-annotator data (Jigsaw, Turkey), we compare the smoothed log-odds used throughout (default), the normalized vote margin $\Delta = (n_{\text{yes}} - n_{\text{no}})/n_{\text{total}}$, and a normalized binary entropy as the cost $|\Delta| = 1 - H(p)/\log 2$. For real-valued data

(NHANES, iNaturalist), we compare the margin $\Delta = z - \tau$ (default) to using the squared costs $|\Delta| = |z - \tau|^2$. Table 3 reports the Standard training NEC under each transform. The choice of transform doesn't change the NEC substantially, except for entropy on Jigsaw. Full per-dataset values and a discussion of robustness are provided in Appendix B.

*Table 3.* NEC (%) of Standard training under alternative $\Delta$ transforms. Default transform is smoothed log-odds for multi-annotator data and raw margin $|\Delta|$ for real-valued data.

| Transform | Jigsaw | Turkey | NHANES | iNat |
|---|---|---|---|---|
| Default | 1.76 | 4.16 | 14.83 | 10.39 |
| Vote margin | 1.63 | 4.60 | — | — |
| Entropy | 0.86 | 3.57 | — | — |
| Squared margin | — | — | 13.47 | 9.27 |

### 4.3. Cost-Sensitive Training

We now evaluate methods that incorporate the per-example costs $|\Delta_i|$ during training: $|\Delta|$-weighted cross-entropy, $\Delta$-regression, and sampling strategies. Across the real datasets, none of these consistently improves NEC over standard cross-entropy training; the one setting where cost-aware training clearly helps is the synthetic dataset. Figure 4 compares the cost-sensitive training methods across datasets, both for NEC and error rate, where the differences are small and inconsistent in direction.

**Fine-tuning.** Figure 5 examines how end-to-end fine-tuning interacts with $|\Delta|$-weighting. The benefit of $|\Delta|$-weighting changes with fine-tuning: on iNaturalist, cost-weighting *hurts* with frozen embeddings (NEC increases from 10.39% to 11.08%) but *helps* with fine-tuning (NEC decreases from 9.34% to 8.77%). This could be explained by fine-tuning learning cost-predictive features that pretrained embeddings lack, enabling cost-weighting to provide benefit. More broadly, the NEC reductions from fine-tuning substantially exceed those from any cost-sensitive training strategy with frozen embeddings.

**Other cost-sensitive methods.** Beyond loss reweighting and sampling, we evaluate four additional approaches from the cost-sensitive learning literature: focal loss (Lin et al., 2017) at $\gamma \in \{1, 2, 3\}$, which downweights confidently-classified examples to focus training on hard ones; *calibrated $|\Delta|$-reweighting* with isotonic regression on the cost magnitudes; *cost-proportionate rejection sampling with ensemble aggregation* following Zadrozny et al. (2003) (**Costing**); and *Bayes-optimal thresholding* following Elkan (2001) (**Elkan threshold**). Table 4 reports NEC for all methods across the five datasets.

No approach consistently improves NEC over Standard CE; every cost-aware method wins on some datasets and loses on

others, with effects rarely exceeding 0.7 percentage points absolute. Calibrated reweighting produces small improvements on several datasets but never by more than 0.6 percentage points, and Costing's effects are mixed in direction. Elkan thresholding is comparable to Standard CE across all datasets (e.g., Jigsaw 1.76% → 1.77%; Turkey 4.16% → 3.67%). $\Delta$-regression is comparable to Standard CE on the real datasets and improves modestly on Synthetic (0.50% → 0.26%). Focal loss is the one method that consistently and substantially harms NEC: at $\gamma = 2$ and $\gamma = 3$ on the real datasets, NEC rises to 82–97%, with the trained model effectively collapsing to predicting one class; even on the synthetic dataset, where costs are linearly predictable from features, focal at $\gamma = 2$ degrades NEC from 0.50% to 17.99%. We attribute this to focal loss downweighting confidently-classified examples, which in our setting are the high-cost (high-$|\Delta|$) examples with strong annotator consensus—the opposite of what NEC rewards.

**Selective classification.** As a complementary view of how cost-weighting affects model behavior at different confidence thresholds, Appendix A reports risk-coverage curves on all five datasets for Standard CE and $|\Delta|$-weighted CE.

## 5. Discussion

### 5.1. Cost Predictability

Cost-weighted training provides inconsistent benefits across datasets. On Turkey, $|\Delta|$-weighting reduces NEC by 14.5%; on iNaturalist, it *increases* NEC by 6.6%; on Jigsaw and NHANES, effects are negligible. The gains (20% reduction) are most noticeable for the synthetic experiment when costs are very predictable from input features.

These findings suggest a candidate principle: cost-sensitive training adds value to the extent that cost magnitudes are predictable from input features. For the synthetic dataset, costs are linearly predictable by construction, and cost-weighting helps. The real datasets, where the feature-cost relationship is weaker, show inconsistent benefits. Additionally, the iNaturalist flip, where cost-weighting hurts with frozen embeddings but helps after fine-tuning, is in line with this principle. Whether this principle generalizes to other instance-reweighting methods is an open question; testing it would require an explicit measure of cost predictability and a broader set of weighting schemes than we evaluate.

### 5.2. Limitations

Our cost derivation assumes that annotator disagreement or distance to threshold correlates with true importance. This assumption need not hold exactly: NEC remains more informative than error rate as long as costs are *ordinally* correct (high-$|\Delta|$ examples genuinely matter more than low-

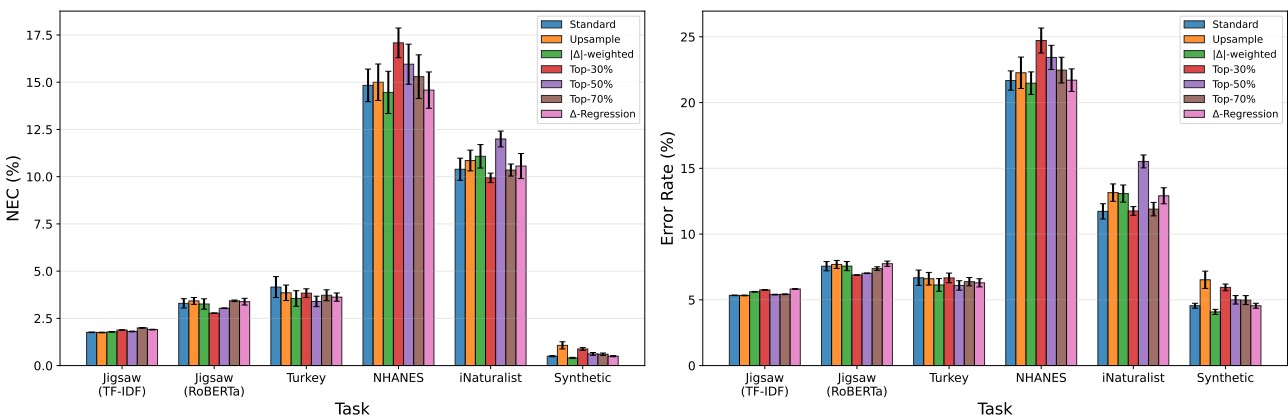

*Figure 4.* Comparison of $\Delta$-based training methods across 6 tasks. Methods include standard training, upsampling, $|\Delta|$-weighting, top-$k\%$ filtering, and $\Delta$-regression.

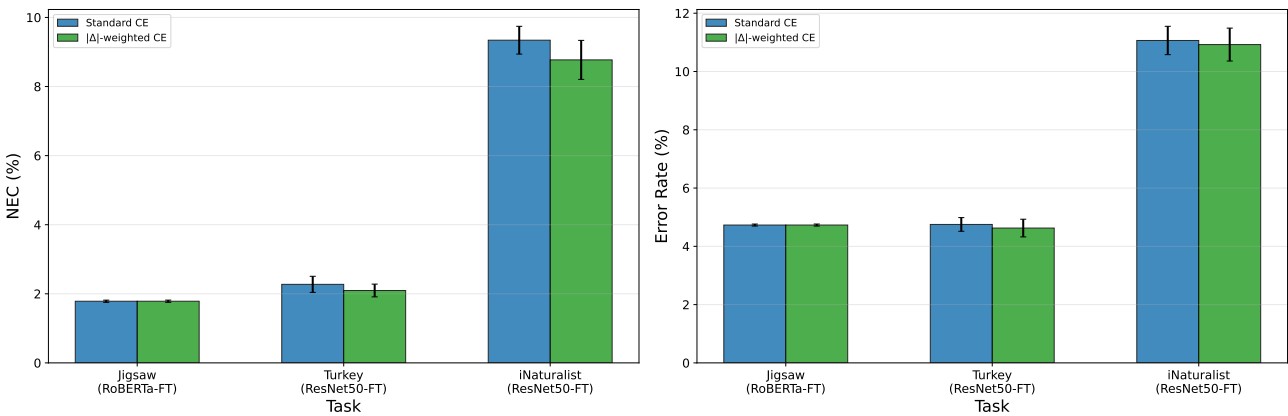

*Figure 5.* Standard vs $|\Delta|$-weighted training for fine-tuned models. Left pane: NEC; right pane: Error Rate. Bar clusters are datasets; colors indicate the training method. Error bars are 95% CI over 10 seeds.

$|\Delta|$ ones). When external costs exist (e.g., user impact, legal liability), these can replace or augment disagreement-derived costs within the same framework.

The iNaturalist dataset relies on LLM-generated confidence ratings rather than human annotator votes. To understand this cost signal, we compared LLM confidence against held-out human judgments on Jigsaw and Turkey samples. Agreement on the binary label is moderate (72.6% on $n$=500 Jigsaw items, 68.0% on $n$=300 Turkey items), but the correlation between LLM-derived $|\Delta|$ and human-derived $|\Delta|$ is weak on Jigsaw (Spearman $\rho = 0.087$, $p = 0.052$) and moderate on Turkey ($\rho = 0.249$, $p < 10^{-5}$). This indicates that LLM confidence captures some but not all of the signal that multi-annotator disagreement captures, and iNaturalist results should be interpreted with this caveat. Where multi-annotator data is available, we recommend it as the primary cost source. Finally, we note that the negative training result is over the cost-sensitive training methods we evaluated ($|\Delta|$-weighted cross-entropy, sampling strategies, $\Delta$-regression, focal loss, calibrated reweighting, Zadrozny costing, and

Elkan thresholding); we cannot rule out improvements from methods we did not test.

## Impact Statement

This paper presents methods for evaluating classifiers using instance-level misclassification costs derived from human disagreement patterns. We highlight several societal considerations.

**Potential benefits.** Our framework encourages practitioners to recognize that not all classification errors are equally consequential. In content moderation, medical screening, and safety-critical systems, this perspective could lead to more nuanced evaluation that better reflects real-world priorities and rewards systems that correctly handle clear-cut cases even if they struggle with genuinely ambiguous ones.

**Potential risks.** Cost derivation from annotator disagreement assumes that consensus reflects importance, which

*Table 4.* NEC (%) for cost-sensitive training methods across datasets. Mean ± 95% CI over 10 seeds. Dashes: see footnote.[†]

| Method | Jigsaw (TF-IDF) | NHANES (HistGBM) | Turkey (ResNet50) | iNaturalist (ResNet50) | Synthetic (LogReg) |
|---|---|---|---|---|---|
| Standard CE | 1.76±0.0 | 14.83±0.9 | 4.16±0.6 | 10.39±0.6 | 0.50±0.0 |
| $|\Delta|$-weighted | 1.79±0.0 | 14.46±1.1 | 3.55±0.4 | 11.08±0.6 | 0.40±0.0 |
| Focal $\gamma = 1$ | 5.13±0.0 | 26.92±1.3 | 7.09±0.4 | 17.99±0.6 | 0.51±0.0 |
| Focal $\gamma = 2$ | 97.34±0.0 | 82.11±0.9 | 95.25±0.3 | 90.26±0.4 | 17.99±1.8 |
| Focal $\gamma = 3$ | 97.19±0.0 | 84.95±0.8 | 94.30±0.5 | 90.24±0.5 | 45.51±2.3 |
| Calibrated reweight | 1.78±0.0 | 14.89±0.7 | 3.61±0.4 | 10.19±0.4 | 0.46±0.0 |
| Costing (Zadrozny)[†] | 2.34±0.0 | 14.48±0.9 | — | — | 0.78±0.0 |
| Elkan threshold | 1.77±0.0 | 15.29±0.7 | 3.67±0.3 | 10.38±0.5 | 0.56±0.0 |
| $\Delta$-regression | 1.91±0.0 | 14.58±1.0 | 3.62±0.2 | 10.56±0.7 | 0.26±0.0 |

[†]Costing was not evaluated on Turkey or iNaturalist due to compute constraints.

may not hold universally. In some domains, unanimous agreement could reflect systematic bias rather than ground truth clarity. Additionally, framing ambiguous cases as "low-cost" could be misused to justify poor performance on examples that are difficult precisely because they affect marginalized groups whose perspectives are underrepresented among annotators. We recommend two mitigations when applying this framework in deployed settings. First, the annotator pool should be diverse along dimensions relevant to the task (demographics, lived experience, domain expertise), and the pool composition should be reported alongside any cost-weighted evaluation. Second, before treating $|\Delta|$ as a cost signal, practitioners should check whether $|\Delta|$ correlates with sensitive attributes or protected-group membership in the data; systematic correlation between low agreement and a particular subgroup is a signal that "ambiguity" may be tracking bias rather than genuine task difficulty.

**Scope.** Our experiments use publicly available datasets for content moderation, medical diagnosis, and species classification. For the iNaturalist dataset, we used LLM-generated annotations to derive costs rather than human annotators. We do not deploy systems in practice. The primary contribution is methodological, by offering a different lens for evaluation, rather than building deployed applications.

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

# A. Selective Classification

Selective classification asks how model performance changes when the system is allowed to abstain from low-confidence predictions (Raghu et al., 2019). At each coverage level $c \in [0, 1]$, we restrict evaluation to the top-$c$ fraction of test examples ranked by predictive confidence $(\max(\hat{p}, 1 - \hat{p}))$ and recompute NEC on this retained subset. Lower NEC at low coverage indicates the model's highest-confidence predictions concentrate on examples it classifies correctly, weighted by cost.

Figure 6 reports selective classification curves on all five datasets, comparing Standard CE against $|\Delta|$-weighted CE. Curves are means over 10 random seeds; shaded bands show 95% CI. At coverage $= 1.0$, Standard CE values match Table 2; $|\Delta|$-weighted values match Table 4.

Across all five datasets, NEC generally decreases as coverage decreases, confirming that confidence-based selective abstention preferentially retains examples on which the model classifies correctly, weighted by cost. The two training methods produce closely-tracked curves on Jigsaw and NHANES, consistent with the negligible-to-modest effect of cost-weighting on these datasets reported in Section 4.3. On Turkey, the two methods cross: $|\Delta|$-weighted CE has higher NEC than Standard CE at low coverage (below $\sim 0.5$) and lower NEC at high coverage (above $\sim 0.7$). iNaturalist shows a different pattern: $|\Delta|$-weighted CE has consistently higher NEC than Standard CE across the full coverage range, with non-overlapping CI bands at higher coverage levels—indicating that the cost-weighting penalty on iNaturalist is not specific to full-coverage evaluation but a consistent feature of how the model ranks examples by confidence. The Synthetic curves stay near zero until coverage $\sim 0.7$ and then rise sharply, and $|\Delta|$-weighted CE is consistently lower than Standard CE on this dataset.

These curves complement the metric-comparison results in Section 4.2: selective classification quantifies the operating-point tradeoff (more abstention $\rightarrow$ lower NEC), while NEC at full coverage is the integrated quantity reported throughout the main paper.

# B. $\Delta$-Transform Sensitivity

The NEC formulation depends on the choice of $\Delta$ transform from raw annotator agreement signals to a per-example cost. Section 4.2 reports the absolute NEC values under each transform; this appendix presents the supporting tables, including rank correlations between configuration orderings.

We evaluate $\Delta$ transforms in two families. For **multi-annotator data** (Jigsaw, Turkey), where $\Delta$ derives from positive vs. negative vote counts, we compare three transforms: smoothed log-odds $\Delta_i = \log \frac{n_{i,\text{yes}}+1}{n_{i,\text{no}}+1}$ (the de-

fault used throughout this paper), normalized vote margin $|n_{i,\text{yes}} - n_{i,\text{no}}|/n_{i,\text{total}}$, and normalized binary entropy $1 - H(p)/\log 2$ where $p$ is the smoothed yes-fraction. For **real-valued data** (NHANES, iNaturalist), where $\Delta$ derives from distance to a threshold or annotator rating, we compare two transforms: the raw margin $|\Delta|$ (default) and the squared margin $\Delta^2$. Entropy is not applicable to real-valued datasets, as they have no vote counts.

For each transform, we report two quantities: absolute NEC of the Standard CE classifier under that transform (averaged over 10 random seeds on the same full-dataset runs as Table 2), and Spearman rank correlation between the configuration ranking under the default transform and the ranking under the alternative (computed across all evaluated configurations on each dataset). Tables 5 and 6 report these for multi-annotator and real-valued datasets, respectively.

*Table 5.* $\Delta$-transform sensitivity on multi-annotator datasets. Absolute NEC (%) is for Standard CE under each transform (matching Table 2 at the default). Spearman $\rho$ is the rank correlation between the configuration ordering under the default transform and the alternative.

| | NEC (%) | | Rank correlation $\rho$ | |
|---|---|---|---|---|
| Transform | Jigsaw | Turkey | Jigsaw | Turkey |
| Default (log-odds) | 1.76 | 4.16 | — | — |
| Normalized margin | 1.63 | 4.60 | 1.00 | 1.00 |
| Entropy | 0.86 | 3.57 | 0.99 | 1.00 |

*Table 6.* $\Delta$-transform sensitivity on real-valued datasets. Absolute NEC (%) is for Standard CE under each transform (matching Table 2 at the default). Spearman $\rho$ is the rank correlation between the configuration ordering under the default transform and the alternative. Entropy is not applicable.

| | NEC (%) | | Rank correlation $\rho$ | |
|---|---|---|---|---|
| Transform | NHANES | iNaturalist | NHANES | iNaturalist |
| Default ($|\Delta|$) | 14.83 | 10.39 | — | — |
| Squared margin | 13.47 | 9.27 | 0.80 | 1.00 |

Across both families, rankings are preserved or near-preserved under all alternative transforms. The one departure is NHANES under the squared margin ($\rho = 0.80$), where squaring amplifies the influence of large-$|\Delta|$ examples (those far from the clinical threshold) and de-emphasizes near-threshold cases. Absolute NEC values shift with the transform—for example, the entropy transform on Jigsaw approximately halves the reported NEC (1.76% $\rightarrow$ 0.86%), reflecting that entropy weights ambiguous cases more strongly than log-odds—but the relative ordering of methods is unchanged. The qualitative conclusions in the main paper do not depend on the transform choice.

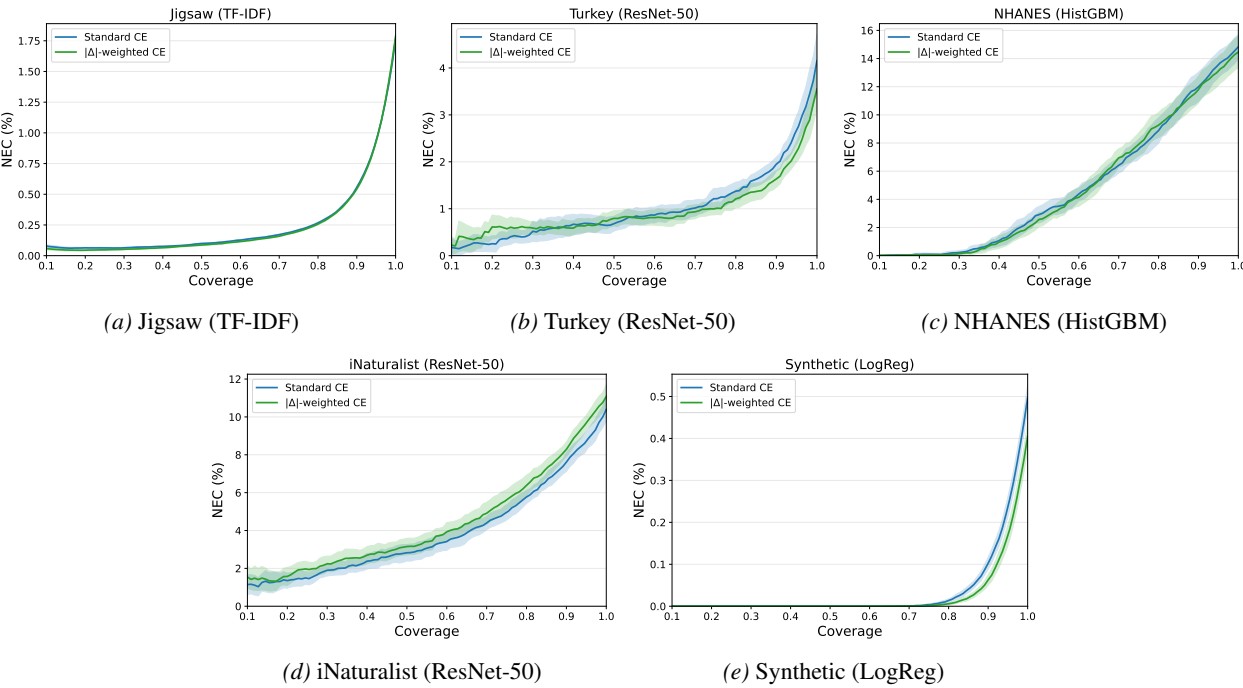

*(a)* Jigsaw (TF-IDF)  *(b)* Turkey (ResNet-50)  *(c)* NHANES (HistGBM)

*(d)* iNaturalist (ResNet-50)  *(e)* Synthetic (LogReg)

*Figure 6.* Selective classification risk-coverage curves across five datasets. Curves are means over 10 random seeds; shaded bands are 95% CI.

