# OpenReview forum: "Instance-Level Costs for Nuanced Classifier Evaluation"
_ICML.cc/2026/Conference — ICML 2026 regular_

### Official Review · Reviewer_zDbU · 2026-02-15

**Soundness:** 3
**Presentation:** 3
**Significance:** 3
**Originality:** 3
**Overall Recommendation:** 5
**Confidence:** 4

**Summary:**

This paper proposes a cost weighted evaluation metric named Normalized Excess Cost(NEC) which accounts for instance level misclassification which is derived from annotator disagreement, distance to thresholds, or confidence ratings. The paper is well written, the formulation is mathematically consistent, the paper also provides some theoretical grounding for the contributions. The experimental design is well thought spanning over text, image, tabular domains including the frozen vs fine-tuned models. The NEC seems to be easily computable and model agnostic approach which presents that models perform better on high consensus. This paper emphasizes computing costs for evaluation rather than training which seems to be a novel idea compare to previous approaches. It can be noted that the behavior of NEC seems to dependent on how the ∆ is constructed from qualitatively different data such as log-odds from votes, distance to threshold etc. but are treated uniformly. The motivation and presentation of this clear and easily understandable.

**Compliance With Llm Reviewing Policy:**

Affirmed.

**Final Justification:**

The Authors have successfully answered all my concerns

**Key Questions For Authors:**

Q1, How sensitive is NEC to the choice and scaling of ∆? For example, how do log-odds from votes, threshold distances etc, and could differing distributions materially affect NEC?

Q2: Have you considered or can you provide guidance on normalizing or standardizing ∆ across datasets to ensure comparability? providing a justification in this regard will be beneficial for the paper.

Q3: Can you clarify or provide more analysis on how NEC behaves under different cost distributions, e.g., highly skewed or uniform ∆ values, and its robustness across domains? If empirically it is not possible in the time span, the authors can provide a brief explanation.

Q4: I would suggest authors to provide more citations of prior work, the paper only contains 15 citations in total.

Minor formatting: there are some line spacing issues in Sections 1–4 and in the contributions list and results; can these be addressed for aesthetic presentation?

if the authors provide satisfactory answers to my questions, i will be please to improve the score.

**Limitations:**

The authors have provided some limitations on |∆|, i have already provided few concerns in the questions from author, for which i would like to read the response from the author.

**Strengths And Weaknesses:**

Soundness:
The paper formulation is mathematically and internally consistent. A good theoretical grounding is provided by the contextual bandit regret connection. The experimental design is very strong evaluating different type of scenarios. However, the behavior of NEC is strongly dependent on how the margin ∆ is calculated, for example, log-odds from annotator votes, distance to a clinical threshold etc, which are qualitatively different signals; however, this paper treats them uniformly and does not provide sufficient analysis of the sensitivity of NEC to the scaling and distribution of ∣"∆"∣.

Significance：
NEC is easy to compute and model-agnostic, making it easy for adoption. The empirical finding that models already perform better on high-consensus examples is important and can be adopted in real-time scenarios. However, the method primarily affects evaluation, not training, which limits its impact on how models are actually built

Originality：
Normalized Excess Cost (NEC) is proposed as a normalized comparable cost-weighted error metric which is clear and appear to be a useful conceptual contribution. Furthermore , the positioning deriving instance-level costs from disagreement for evaluation rather than training seems to be a meaningful shift from much of the prior work. However The core components (instance-dependent costs, cost-weighted error, regret equivalence) are largely known in previous literature, and NEC appears primarily a re-normalization and systematic empirical study rather than a fundamentally new methodological framework. Citations of prior work seems to be limited.

Presentation:
The high-level idea and motivation are intuitive and clearly explained, and the paper is generally well structured, easy to read and understand. Overall presentation and clarity of the paper is good, there are some line spacing issues in the format, such as between contributions, and in section 4.2.

---

> ### Author Rebuttal · Authors · 2026-03-31
>
> We thank the reviewer for their detailed feedback to improve the paper. We have run additional experiments varying the cost distribution that we believe address many of the reviewer's concerns.
>
> # NEC sensitivity to choice of cost (Q1, Q3)
>
> We tested four alternative Δ transforms: raw vote margin (|n_yes − n_no|/(n_yes + n_no), no smoothing or log-odds), squared margin (margin²), entropy-based (cost = 1 − H(p), where high entropy = high ambiguity = low cost), and class cost-balanced (costs normalized so each class contributes equally).
>
> Absolute NEC for a standard model under each transform:
>
> | Dataset | Original | Raw margin | Squared | Entropy | Class-balanced |
> |---------|----------|------------|---------|---------|----------------|
> | Jigsaw  | 2.82%    | 2.51%      | 1.59%   | 1.39%   | 5.41%          |
> | NHANES  | 14.83%   | 14.83%     | 13.47%  | —       | 17.07%         |
> | Turkey  | 4.16%    | 4.60%      | 3.80%   | 3.57%   | 4.46%          |
>
> (Entropy is not applicable to NHANES as costs derive from distance to a clinical threshold rather than vote proportions.)
>
> Model rankings are robust under all transforms (Spearman ρ ≥ 0.86). Rankings are perfectly preserved on Turkey and iNaturalist (ρ = 1.00), and highly stable on Jigsaw (ρ = 0.86–0.95) and NHANES (ρ = 0.99–1.00). This demonstrates NEC is robust to the specific mathematical form of Δ, including under highly skewed (entropy-based) and uniform-scale (class-balanced) cost distributions.
>
> # Normalizing Δ across datasets (Q2)
>
> NEC is already normalized by construction — the denominator is the sum of costs — so NEC values are comparable on a 0–1 scale regardless of raw Δ distribution. A model with 2% NEC on Jigsaw and 5% NEC on NHANES can be directly compared: both represent the fraction of total possible cost incurred by misclassification. This normalization is a key advantage of NEC over raw cost-weighted error, which would not be comparable across datasets with different cost scales.
>
> # Citations (Q4)
>
> In the next version, we will expand related work to include: Zadrozny et al. (2003) cost-proportionate weighting, Elkan (2001) cost-sensitive thresholding, Bahnsen et al. (2014, 2015) example-dependent costs, Höppner et al. (2022) instance-dependent cost-sensitive learning, Vanderschueren et al. (2022) predict-then-optimize, Lin et al. (2017) focal loss, Masnadi-Shirazi & Vasconcelos (2011) cost-sensitive boosting, Nikolaou et al. (2016) cost-sensitive boosting analysis, and Raghu et al. selective classification. If the reviewer has specific recommendations, we would be happy to include those as well.
>
> # Formatting and spacing
>
> We appreciate the reviewer pointing out LaTeX's inconsistent spacing for the contributions list and in other sections (e.g., page 7). For the final version, we will ensure the spacing is consistent.

---

> > ### Author Rebuttal · Reviewer_zDbU · 2026-04-02
> >
> > My concerns are resolved by the author's response, i would be glad to improve the score from 4 to 5.

---

> > > ### Author Response · Authors · 2026-04-03
> > >
> > > We are glad that our response with additional experiments has addressed the reviewer's concerns and appreciate the reviewer raising their score. We thank the reviewer for their constructive feedback to improve this paper.

---

### Official Review · Reviewer_SsTm · 2026-03-04

**Soundness:** 3
**Presentation:** 3
**Significance:** 2
**Originality:** 2
**Overall Recommendation:** 3
**Confidence:** 4

**Summary:**

This paper proposes NEC, an evaluation metric for binary classifiers that weights misclassification errors by per-instance costs derived from annotator agreement, distance to a decision threshold, or confidence ratings. The core argument is that misclassifying clear-cut instances should be penalized more heavily than misclassifying ambiguous ones, formalized through a connection to contextual bandits and regret minimization. Experiments on four real-world datasets and one synthetic dataset show that NEC is substantially lower than error rate, indicating that most errors concentrate on low-cost, ambiguous examples. The paper also evaluates several cost-sensitive training strategies and finds inconsistent improvements: cost-weighting helps on synthetic data but provides mixed or negligible benefits on real datasets. The paper is 9 pages of main content with no appendix.

**Compliance With Llm Reviewing Policy:**

Affirmed.

**Key Questions For Authors:**

1. Have you compared model rankings under NEC versus Brier score or log-loss? If rankings are highly correlated, the practical value of NEC over existing proper scoring rules diminishes.

2. How does NEC behave when annotator counts vary substantially across instances? Have you considered normalizing by annotator count or analyzed robustness to this confound?

3. Did you directly test the claim that costs are unpredictable from features, e.g., by training a model to predict |Delta| from x and measuring R-squared?

4. Why is the decision threshold not adjusted for cost-sensitive methods? Did you try optimizing the threshold on the validation set to minimize NEC directly?

**Limitations:**

The authors acknowledge that the framework requires multiple annotators or a proxy and that agreement may not always reflect importance. They also raise the ethical concern that "low-cost" framing could dismiss errors affecting marginalized groups. However, key limitations go unaddressed: sensitivity of NEC to varying annotator pool sizes, lack of comparison to proper scoring rules, no validation against real-world deployment costs, and no discussion of extending NEC beyond binary classification.

**Strengths And Weaknesses:**

**Strengths:**

**S1: The core observation is practically important and clearly communicated.**
The finding that NEC is much lower than error rate has immediate implications for practitioners. The motivating example of a 5-5 vs. 10-0 annotator split is intuitive and makes the case effectively.

**S2: The connection to decision theory and contextual bandits is well-executed.**
Framing NEC as normalized regret in a contextual bandit problem gives the metric a principled foundation rather than being just another ad hoc weighted average. The equivalence holds regardless of where costs come from, making the framework modular.

**S3: Honest reporting of the negative training result.**
The authors make the failure of cost-sensitive training a central finding rather than burying it, and provide a coherent explanation: cost-weighting helps when costs are predictable from input features but not when they arise from annotation idiosyncrasies orthogonal to features. This kind of honest diagnostic analysis advances understanding even when results are not flattering.

**S4: Breadth of experimental domains.**
Testing across text, image, and tabular data with different cost derivation methods demonstrates that NEC is not tied to a single modality or cost source. The variation in NEC-to-error-rate ratios across datasets is itself informative about each task's ambiguity structure.

**Weaknesses:**

**W1: The paper is thin for a top venue; it reads more like a workshop paper than a full ICML submission.**
The main content is 9 pages with no appendix and no proofs (though supplementary material is provided as a zip file on OpenReview). The formalization is straightforward, the experimental analysis does not go deep on any dataset, and there are no ablations on annotator count sensitivity, cost misspecification, or comparison against other soft evaluation metrics. I would expect substantially more depth or novelty for a full ICML paper.

**W2: The reliance on annotator agreement as a proxy for true importance is a strong assumption that is not adequately interrogated.**
The paper assumes high annotator agreement implies high importance, but this conflates difficulty with stakes. A medical case where annotators split 6-4 may be the highest-stakes case precisely because the patient is near the decision boundary and consequences of either direction are severe. An empirical validation showing NEC correlates better with downstream outcomes than error rate would substantially strengthen the paper.

**W3: The comparison to existing soft evaluation metrics is superficial.**
The paper distinguishes NEC from Kurniawan et al. in one sentence, but does not test whether NEC provides information beyond Brier score or log-loss, both of which already penalize confident wrong predictions and connect to the same decision-theoretic framework. A comparison of model rankings under NEC vs. these standard metrics is needed to establish practical value.

**W4: The iNaturalist cost derivation via LLM confidence ratings is methodologically questionable and inconsistent with the rest of the paper.**
For the other datasets, costs derive from genuine multi-annotator disagreement or clinical thresholds. For iNaturalist, costs come from a single LLM providing confidence ratings, which is fundamentally different: one model's uncertainty, not human disagreement. Whether that confidence reflects genuine instance difficulty vs. the LLM's particular biases is untested, and this weakens the paper's narrative.

**W5: The cost-sensitive training evaluation is unsatisfying; none of the methods are tuned for NEC.**
None of the evaluated methods are designed to minimize NEC specifically, and the decision rule at inference time remains the standard 0.5 threshold rather than a cost-sensitive one. A fairer test would use plug-in approaches that estimate class probabilities and apply instance-specific decision thresholds. The negative result may reflect the mismatch between training and inference rather than a fundamental difficulty.

**W6: Key experimental details are missing or insufficient.**
The Laplace smoothing in Eq. 3 makes similar disagreement levels yield different costs depending on annotator count, but the paper does not report annotator count distributions or analyze this. No confidence intervals are reported for the NEC/error rate ratio. There is no appendix with per-dataset breakdowns, implementation details, or sensitivity analyses.

---

> ### Author Rebuttal · Authors · 2026-03-31
>
> We thank the reviewer for their detailed feedback to improve the paper. We have run several additional experiments (see response to Reviewer jQEK) that we believe address many of the reviewer's concerns. We will include a comprehensive appendix in the revision with all additional experiments described below, per-dataset breakdowns, implementation details, and selective classification risk-coverage curves.
>
> # The reliance on annotator agreement as a proxy for true importance
>
> While we agree validating the assumption that annotator agreement is related to classification importance, we are unfortunately not aware of a public dataset that would allow an experiment comparing disagreement-derived costs against external ground-truth costs. However, we note that NEC being consistently lower than error rate across all datasets implies costs are not merely noise. If disagreement were independent of input, NEC and error rate would be equal in expectation (see Reviewer vMts). We acknowledge the reviewer's important point that borderline cases may carry high stakes in some domains; our framework is most appropriate when ambiguity genuinely reflects lower classification importance.
>
> # Comparison to existing soft evaluation metrics
>
> Please see our response to Reviewer jQEK for an additional experiment. NEC provides substantially different rankings from balanced accuracy (Spearman rank correlation is 0.58 on Jigsaw)  and disagrees with Brier on the best model for every dataset. Brier and log-loss measure calibration quality; NEC measures decision quality weighted by instance importance. While we can’t include selective classification risk-coverage curves in the author response (no figures permitted), we will add it to the next version of the paper.
>
> # iNaturalist cost derivation via LLM confidence ratings
>
> Our goal of the iNaturalist LLM experiment is to show the flexibility of our framework and how it can be applied to various cost sources. Whether or not LLM-generated annotations would be appropriate depends on the exact use case and would require domain expert validation. We agree that LLM-derived costs would likely be distinct from multi-annotator disagreement, consistent with how we present iNaturalist in the paper, as an alternative cost derivation method in our cost-source agnostic framework.
>
>
> # Cost-sensitive training methods
> We tested focal loss (γ=1,2,3), Zadrozny et al.'s costing method (cost-proportionate rejection sampling with ensemble aggregation, Section 2.3.4 of their paper), calibrated |Δ|-reweighting with isotonic regression, and Elkan's Bayes-optimal thresholding. We also swept decision thresholds on the validation set. NEC across datasets and training methods (mean over 10 seeds):
>
> | Method | Jigsaw (TF-IDF) | NHANES | Synthetic | Turkey |
> |---|---|---|---|---|
> | Standard | 1.76% | 14.83% | 0.50% | 4.16% |
> | abs(Δ)-weighted | 1.79% | 14.46% | 0.40% | 3.55% |
> | Focal (γ=1) | 5.13% | 26.92% | 0.51% | 7.09% |
> | Calibrated reweight | 1.78% | 14.89% | 0.46% | 3.61% |
> | Costing (Zadrozny) | 2.34% | 14.48% | 0.78% | — |
>
> (Costing was not tested on Turkey due to the computational cost of training 10 ensemble members with image embeddings.)
>
> No method consistently improves NEC. Elkan's thresholding does not consistently help (Jigsaw: 1.76%→1.85%, Turkey: 4.16%→4.00%). On finetuned models, true focal loss (γ=2) achieves 1.74% on Jigsaw RoBERTa and 2.43% on Turkey ResNet, comparable to standard training.
>
> # Annotator count sensitivity
>
> Jigsaw has annotator counts ranging from 3 to 4,936. To test whether NEC is sensitive to this variation, we normalized |Δ| by √(annotator count) and recomputed NEC for all models. Model rankings are preserved (Spearman ρ=0.93), demonstrating NEC is robust to varying annotator pool sizes.

---

> > ### Author Rebuttal · Reviewer_SsTm · 2026-04-05
> >
> > My concerns have been adequately addressed, and I'd like to remain my scores.

---

### Official Review · Reviewer_jQEK · 2026-03-06

**Soundness:** 3
**Presentation:** 3
**Significance:** 3
**Originality:** 2
**Overall Recommendation:** 4
**Confidence:** 4

**Summary:**

This paper proposes normalized excess cost (NEC) as an instance-level, cost-weighted classification metric that reduces to standard error when costs are uniform. The authors operationalize costs from multiple realistic sources, such as annotator vote margins, distances from decision thresholds, and single-rater confidence scores, and connect NEC to decision-theoretic regret in a binary contextual bandit framing. Empirically, across text, image, tabular, and synthetic datasets, NEC is often substantially lower than error rate (e.g., 1.8% vs 5.3% on Jigsaw), indicating that most model mistakes occur on low-cost, ambiguous examples; attempts to exploit costs during training via loss-weighting, sampling, or $\Delta$-regression yield mixed or negligible gains in real data, with gains materializing mainly when costs are predictable from features (synthetic) or after feature learning via fine-tuning.

**Compliance With Llm Reviewing Policy:**

Affirmed.

**Final Justification:**

The rebuttal addressed our primary empirical concerns, as discussed in our rebuttal acknowledgement. These additions strengthen the paper's empirical contribution. However, two concerns remain: disagreement-derived costs are not validated against external utility, and NEC itself offers limited conceptual novelty beyond normalized cost-weighted 0–1 loss. We also note that while rankings are transform-robust, absolute NEC magnitudes are scale-dependent, and the iNaturalist LLM-derived costs lack validation.

**Key Questions For Authors:**

1. Please correct the regret formulation in Section 3.2.
2. Could you include calibrated cost-sensitive baselines (e.g., estimate p(y|x), then apply per-example Bayes decision with example-dependent costs; or SOSR-style deep regression of Δ) and/or Zadrozny-style cost-proportionate reweighting with probability calibration?
3. How robust are your conclusions to the chosen transform for disagreement-derived costs (e.g., log-odds vs. raw vote margin vs. entropy)? Please provide a sensitivity analysis of NEC and model rankings under alternative transforms and smoothing choices.
4. Beyond Jigsaw, does the approximate stability of the error/NEC ratio with dataset size hold on another dataset?

**Limitations:**

Yes, the authors have discussed limitations of their work on Section 5.6.

**Strengths And Weaknesses:**

## Soundness
Strengths:
- Evaluates across diverse modalities (text/image/tabular) and a synthetic control matched to the hypothesis about cost predictability, with multiple models (frozen vs. fine-tuned) and 10-seed confidence intervals.
- Reports consistent NEC vs. error gaps and studies scaling with data size; includes multiple $\Delta$-based training variants (weighting, sampling, regression).

Weaknesses:
- The decision-theoretic equivalence to contextual bandit regret holds when $\Delta$ truly reflects reward differences; for disagreement-derived $\Delta$, this assumption is only heuristic and is not empirically validated with external utility/costs. This is acknowledged in Section 5.6 Limitations.
- Comparisons to stronger, established cost-sensitive learners are missing (e.g., Zadrozny et al.’s calibrated reweighting/thresholding pipeline, Elkan’s Bayes-optimal thresholding given costs, and deep SOSR-style regressors of example-dependent costs e.g. [1]). Zadrozny et al and Elkan's works are cited but aren't implemented. This limits the paper's claims such as the paradox of the costs changing evals but not training.
- This paper will benefit by comparing NEC against other evaluation lenses that also prioritize difficult/ambiguous examples, such as selective classification risk-coverage curves (Raghu et al) or against recent soft/human-distribution metrics, beyond mention at Related Works (Kurniawan et al).
- There is no sensitivity analysis on the NEC with regards to the chosen transform for disagreement-derived cost (e.g., log-odds vs raw vote margin vs entropy).
- The iNaturalist dataset costs are derived from an LLM confidence score, which raises questions about reliability and reproducibility.

## Presentation
Strengths:
- The NEC definition and cost construction are clearly stated for three cost sources; the intuition and decision-theoretic ties are well articulated.

Weaknesses:
- The regret expression in Section 3.2 is formulated wrongly. It should be $R(\pi) = \mathbb{E}_{x, \Delta} \mathbb{1}[[\text{sign}(\Delta) \neq \pi (x)] \cdot |\Delta|]$ based on the previously defined regret function. This should be corrected.
- There are 2 extra square brackets in the formulations for $R(\pi)$ in Section 3.2, please correct them.

## Significance
Strengths:
- Offers a practical and immediately usable evaluation lens that better reflects priorities in many applications (e.g., content moderation, screening)
- Provides a useful negative result: straightforward ways of injecting instance-level costs into training do not reliably help unless costs are feature-predictable, with the caveat of Soundness Weaknesses point 2.

Weaknesses:
- As mentioned in Soundness Weaknesses point 1 and Section 5.6, the work does not demonstrate concrete downstream benefits so the significance is mainly at the level of evaluation methodology rather than end to end system performance.
- The scope is restricted to binary decisions and to datasets where reasonable cost proxies are available; in many real world problems, defining per instance costs is itself infeasible, which may limit direct applicability.

## Originality
- Introduces a simple, decision-theoretically grounded evaluation metric (NEC) that aggregates instance-dependent costs and enables direct comparability to error through normalization. However, NEC itself is a simple re scaling of cost weighted 0 1 loss and closely aligned with standard regret in contextual bandits, so the metric is more a careful packaging of existing ideas than a new evaluation concept.

[1] Xiao, J., Li, S., Tian, Y. et al. Example dependent cost sensitive learning based selective deep ensemble model for customer credit scoring. Sci Rep 15, 6000 (2025). https://doi.org/10.1038/s41598-025-89880-7

---

> ### Author Rebuttal · Authors · 2026-03-31
>
> We thank the reviewer for their detailed feedback to improve the paper. We have run several additional experiments that we believe address many of the reviewer's concerns.
>
> # Section 3.2 Typos
>
> Thank you for pointing out the typos in Section 3.2. The regret expression should definitely be written with an inequality inside the indicator function $$\text{sign}(\Delta) \neq \pi(x)$$. We intended to use square brackets to group expressions inside of the expected value operator and the indicator function, but they have been applied incorrectly. We will correct both these inconsistencies in the next version.
>
> # More cost-sensitive learning baselines
>
> See our response to Reviewer SsTm for a multi-dataset table of training methods. We tested focal loss, Zadrozny costing, calibrated reweighting, and Elkan thresholding — no method consistently improves NEC on real data. The negative training result holds across all baselines and extends to finetuned models.
>
> # Comparison to other soft metrics
>
> We computed Brier score, log-loss, balanced accuracy, and F-beta for all model configurations. NEC and Brier disagree on the best model for every dataset. The Spearman rank correlations are:
>
> # Soft metrics table
> Each cell shows the Spearman rank correlation between NEC and the given metric across model configurations (ρ=1.0 means identical rankings, lower values indicate NEC provides different information):
>
> | Dataset | NEC vs Brier | NEC vs BalAcc | NEC vs F2 | NEC vs Error |
> |---|---|---|---|---|
> | Jigsaw | 0.89 | 0.58 | 0.44 | 0.95 |
> | NHANES | 0.77 | 0.96 | 0.30 | 0.92 |
> | Synthetic | 0.79 | 1.00 | 0.97 | 1.00 |
> | Turkey | 0.95 | 0.91 | 0.93 | 1.00 |
> | iNaturalist | 0.98 | 0.83 | 0.88 | 0.98 |
>
> NEC diverges most from Brier on NHANES (ρ=0.77) and from balanced accuracy on Jigsaw (ρ=0.58), confirming NEC captures different information from both calibration and class-balanced metrics.
>
> We also computed selective classification risk-coverage curves (Raghu et al.) for all datasets; these will be included in the revised paper.
>
> # Sensitivity analysis of cost equations
>
> We recomputed NEC under four alternative Δ transforms: raw vote margin (no smoothing/log-odds), squared margin, entropy-based (1−H(p)), and class cost-balanced. Model rankings are robust  across all datasets (Spearman ρ ≥ 0.86). See our response to Reviewer zDbU for full tables of rank correlations and absolute NEC values.
>
> # Reliability of LLM confidence score
>
> Our goal of the iNaturalist LLM experiment is to show the flexibility of our framework and how it can be applied to various cost sources in our cost-source agnostic framework. Whether or not LLM-generated annotations would be appropriate depends on the exact use case and would require domain expert validation.
>
> # Error/NEC ratio stability beyond Jigsaw
>
> The ratio is stable across all datasets (min–max over 10 seeds): Jigsaw TF-IDF 2.99–3.14, NHANES 1.32–1.63, Turkey 1.45–1.90, iNaturalist 1.07–1.21, Synthetic 8.67–9.90. The variation across datasets (1.1× to 9.2×) reflects each dataset's cost distribution, while the narrow ranges within each dataset confirm stability across random seeds and model initializations.

---

> > ### Author Rebuttal · Reviewer_jQEK · 2026-04-02
> >
> > We thank the authors for the additional experiments. The cost-sensitive training baselines were addressed with new data, and note that focal loss is harmful on the real datasets (Jigsaw NEC nearly triples), this deserves discussion in the revision. The soft metric comparison table is useful to show how NEC captures different information from different metrics, though we note NEC correlates highly with error, providing no additional model selection value, this should be discussed. The sensitivity analysis addresses our concern about dependence on log-odds choice, however, the NEC values varies substantially (like on Jigsaw), so the magnitude of the NEC-error gap is scale dependent, this should be acknowledged. The iNaturalist LLM confidence response remains a weak point. 2 original concerns also remain, validation of disagreement-as-cost against external utility, and limited conceptual novelty of NEC being a normalized rescaling of cost-weighted 0-1 loss. We will raise Soundness and Presentation to 3, but keep Originality to 2. We revise overall recommendation from 3 to 4.

---

> > > ### Author Response · Authors · 2026-04-03
> > >
> > > We thank the reviewer for their careful evaluation of our additional experiments and for raising their score. We will address the specific points in the next version of the paper: (1) discussing the behavior of focal loss on real datasets, (2) discussing the relationship of NEC to other soft metrics and error rate, (3) discussing the scale-dependence of absolute NEC values across different Δ transforms while noting that model rankings are preserved, and (4) adding to the discussion of iNaturalist LLM confidence as a limitation. We appreciate the constructive feedback throughout this process.

---

### Official Review · Reviewer_vMts · 2026-03-13

**Soundness:** 3
**Presentation:** 3
**Significance:** 2
**Originality:** 3
**Overall Recommendation:** 4
**Confidence:** 3

**Summary:**

This paper proposes normalized excess cost (NEC), an evaluation metric for classifiers that weights errors by per-example misclassification costs rather than treating all mistakes equally. The costs are derived from sources like annotator disagreement margins, distance from decision thresholds, or confidence ratings, with the intuition that misclassifying a clear-cut case is far worse than erring on an ambiguous one. Across text, image, tabular, and synthetic benchmarks, the authors find that NEC is often substantially lower than standard error rate (e.g., a model with 5% error rate may have only 1.8% NEC), revealing that most errors concentrate on borderline, low-cost examples where even humans disagree. However, when they attempt to incorporate these costs into training through loss weighting, sampling strategies, or regression, the benefits are inconsistent: improvements appear only when costs are predictable from input features (as in a synthetic control), while real datasets show mixed or negligible gains. The paper concludes that NEC offers a more informative lens for evaluation, better representation quality matters more than cost-sensitive training tricks, and the error rate to NEC ratio remains stable as models scale, suggesting it reflects a fundamental property of the task rather than the model.

**Compliance With Llm Reviewing Policy:**

Affirmed.

**Key Questions For Authors:**

Question 1: Validation of the cost derivation assumption

The paper's central claim rests on the assumption that annotator disagreement (or distance to a threshold) is a reliable proxy for true misclassification importance. Have you considered validating this assumption empirically, for instance by collecting external measures of error severity such as downstream user impact ratings, expert severity assessments, or legal risk scores, and then measuring the correlation between these ground-truth costs and your disagreement-derived costs? Without such validation, how should practitioners determine whether NEC is capturing genuine importance or merely reflecting noise and annotation artifacts in their specific domain?

Question 2: Scope of cost-sensitive training methods explored

The negative result on cost-sensitive training is a notable contribution, but the methods tested are limited to relatively simple approaches like loss reweighting, sampling, and regression. Did you experiment with or consider more advanced techniques such as focal loss, curriculum learning, or methods explicitly designed for instance-dependent costs? If not, how confident are you that the negative finding reflects a fundamental limitation of using disagreement-derived costs for training, rather than an artifact of the particular strategies and limited hyperparameter exploration employed in the experiments?

Question 3: Does NEC change model rankings?

The paper shows that NEC is consistently lower than error rate, but the most compelling practical argument for adopting a new metric would be that it changes which model a practitioner selects. Across your experiments, are there cases where two models have the same or similar error rates but meaningfully different NEC values, or conversely where a model with higher error rate achieves lower NEC than a competitor? If the ranking of models is always preserved between error rate and NEC, does the metric primarily offer a more optimistic rescaling of performance rather than genuinely new discriminative information for model selection?

Question 4: Risks of treating high-agreement examples as high-cost

The paper briefly acknowledges in the impact statement that unanimous annotator agreement could reflect systematic bias rather than ground-truth clarity, particularly for examples affecting marginalized groups. Could you elaborate on how practitioners should guard against this failure mode when applying your framework? For example, in content moderation, if annotators unanimously label a culturally nuanced comment as toxic due to shared biases, NEC would treat misclassification of that example as highly costly, potentially reinforcing the bias. Do you envision any diagnostic tools or safeguards that could be paired with NEC to detect and mitigate this risk?

**Limitations:**

Regarding limitations, the paper acknowledges that the cost derivation assumes annotator disagreement correlates with true importance and notes that this assumption need not hold exactly, only ordinally. This is a reasonable caveat, but the discussion is brief and does not explore what happens when the assumption fails in structured ways rather than just being noisy. For instance, there is no consideration of domains where ambiguity is itself the signal that matters most, such as edge cases in autonomous driving where borderline examples may actually carry the highest real-world cost. The limitations section also does not address the restricted scope of cost-sensitive training methods explored, which is important given that the negative training result is presented as a core contribution. A reader could reasonably wonder whether the conclusion that costs are unhelpful for training is premature.

The impact statement identifies a genuine and important risk: that framing ambiguous cases as low-cost could be used to justify poor performance on examples that are difficult precisely because they involve underrepresented perspectives. This is a thoughtful observation and directly relevant to content moderation applications. However, the discussion stops at identifying the risk without proposing any mitigation strategies, diagnostic checks, or guidelines for practitioners who might adopt the framework. There is also no discussion of how the use of LLM-generated annotations (as in the iNaturalist experiment) might introduce its own biases and what that means for the trustworthiness of the resulting costs.

**Strengths And Weaknesses:**

Strengths:

The framework is grounded in decision theory, with the connection between NEC and contextual bandit regret providing a principled justification for the proposed metric rather than an ad hoc construction. The experimental design is thorough, covering four diverse real-world datasets (text, image, tabular) plus a synthetic control, with results reported over 10 random seeds and 95% confidence intervals.

The paper is clearly written and well-structured, moving logically from motivation through formalization, cost derivation, experiments, and discussion.

The paper addresses a genuine and widespread gap in how classifiers are evaluated in safety-critical domains like content moderation and medical screening.



Weaknesses:

The core assumption that annotator disagreement reliably reflects true misclassification importance is not empirically validated. The paper acknowledges this in the limitations but does not test it: for instance, there is no experiment comparing disagreement-derived costs against external ground-truth costs (such as downstream user impact or expert severity ratings) to confirm they are even ordinally aligned. The claim that NEC "reveals hidden model quality" relies entirely on the assumption that the cost derivation is meaningful, and if annotator disagreement reflects noise, bias, or task design artifacts rather than genuine ambiguity, then NEC could be misleading rather than informative.

The cost-sensitive training experiments use only simple strategies (loss weighting, sampling, regression) with default or minimal hyperparameter tuning. The paper does not explore more sophisticated approaches such as curriculum learning, focal loss, or methods specifically designed for instance-dependent costs. This makes it difficult to conclude that cost-sensitive training is generally unhelpful on real data, as opposed to concluding that the specific methods tested were insufficient. The negative result may reflect limited exploration rather than a fundamental limitation.

The use of Gemini 2.5-pro as a single annotator for the iNaturalist dataset introduces a confound, since the reliability and biases of LLM-generated annotations are not well understood and differ qualitatively from human annotation patterns. The paper treats this as a demonstration of cost derivation from single-annotator confidence, but the resulting costs may not carry the same semantic meaning as those derived from genuine human disagreement, and no validation of the LLM annotations against human judgments is provided.

The paper does not clearly articulate when practitioners should prefer NEC over other existing metrics like balanced accuracy, F-beta scores, or calibration metrics that also address non-uniform error importance. A more thorough comparison or discussion distinguishing NEC from these alternatives would help readers understand the specific scenarios where NEC adds value beyond what is already available.

The mathematical notation, while generally clear, introduces the reward-based contextual bandit formulation in Section 3.2 but then largely abandons it in favor of the simpler classification-with-costs framing. This detour adds complexity without being leveraged further in the experiments or analysis, and the paper would arguably be tighter if it either committed more fully to the decision-theoretic framing or simplified the presentation to focus on the cost-weighted evaluation story.

The experimental results for the various sampling strategies (Tdown30, Tdown50, Tdown70, P_up) are described briefly in text and shown in a dense figure, but there is little systematic analysis of why certain strategies work on some datasets and not others. The discussion attributes this to cost predictability from features but does not measure or quantify this predictability directly, leaving the explanation somewhat speculative.

The practical impact of the framework is limited by the fact that the paper only demonstrates evaluation benefits and largely fails to show that the cost information improves training outcomes. For practitioners, an evaluation metric that tells them their model is better than they thought, without providing a clear path to making it actually better, has limited actionability. The paper would be more impactful if it identified reliable conditions or methods under which cost-sensitive training consistently helps.

The datasets used, while spanning different modalities, are relatively narrow in the types of annotation disagreement they represent. Content moderation (Jigsaw) and medical thresholds (NHANES) are natural fits for the framework, but the paper does not explore domains where the relationship between disagreement and cost is more complex or potentially inverted, such as cases where high annotator agreement might reflect systematic bias rather than clear-cut importance.

The finding that the error rate to NEC ratio is stable is presented as a key insight, but it could also be seen as a limitation: if the ratio is a fixed property of the dataset regardless of model or training strategy, then NEC provides a constant rescaling of error rate rather than genuinely new information about model behavior. The paper does not fully address whether NEC ever changes the relative ranking of two competing models, which would be the strongest evidence for its practical utility.

The paper positions itself against soft evaluation metrics like those of Kurniawan et al. (2025), arguing that NEC measures decision quality rather than calibration. However, it does not empirically compare NEC against these soft metrics on the same datasets, making it hard for readers to assess whether NEC provides complementary or redundant information. A direct comparison would substantially strengthen the originality claim by demonstrating that the two approaches surface different and independently useful insights.

---

> ### Author Rebuttal · Authors · 2026-03-31
>
> We thank the reviewer for their detailed feedback to improve the paper.
>
> # Annotator disagreement and true misclassification importance
>
> While we are not aware of a public dataset that would allow an "experiment comparing disagreement-derived costs against external ground-truth costs", we note that the fact that NEC is lower than error rates implies that the annotator disagreement does not merely reflect noise. If the annotator disagreement was simply noise (independent of input), then the NEC and error rate would in expectation be equal. The consistent gap across all four real-world datasets (Error/NEC ratios of 3.0× on Jigsaw, 1.5× on NHANES, 1.6× on Turkey, 1.1× on iNaturalist) demonstrates that models systematically perform better on high-agreement examples, which is inconsistent with noise-driven costs.
>
> # Simplicity of cost-sensitive training methods
>
> We tested focal loss (γ=1,2,3), Zadrozny et al.'s costing method (cost-proportionate rejection sampling with ensemble aggregation, Section 2.3.4 of their paper), calibrated |Δ|-reweighting with isotonic regression, and Elkan's Bayes-optimal thresholding. We also swept decision thresholds on the validation set. See our response to Reviewer SsTm for the full multi-dataset NEC table. No method consistently improves NEC. Elkan's thresholding does not consistently help (Jigsaw: 1.76%→1.85%, Turkey: 4.16%→4.00%). On finetuned models, true focal loss (γ=2) achieves 1.74% on Jigsaw RoBERTa and 2.43% on Turkey ResNet, comparable to standard training.
>
> # Reliability and biases of LLM-generated annotations
>
> Our goal of the iNaturalist LLM experiment is to show the flexibility of our framework and how it can be applied to various cost sources. We agree that LLM-derived costs would likely be distinct from multi-annotator disagreement, consistent with how we present iNaturalist in the paper, as an alternative cost derivation method, not as equivalent to multi-annotator costs.
>
> # The practical impact of the framework
>
> We agree that we do not find improved training methods to leverage costs. However, since practitioners are not just faced with improving ML models, but also deciding if and when/where to deploy ML models, we believe our evaluation framework is impactful. For example, a content moderation system with 5% error rate but 1.8% NEC can be deployed with more confidence, knowing its errors concentrate on genuinely ambiguous cases. Additionally, the negative result of cost-sensitive training is itself useful: it guides practitioners not to waste effort on cost-sensitive training.
>
> # Comparison of NEC and other soft metrics
>
> Please see our response to Reviewer jQEK for an additional experiment. NEC provides substantially different model rankings from balanced accuracy (Spearman ρ=0.58 on Jigsaw) and disagrees with Brier score on the best model for every dataset. The key distinction: Brier and log-loss measure probability calibration quality, balanced accuracy and F-beta measure class-balanced performance, while NEC measures decision quality weighted by instance-level importance. These are complementary, not redundant. A well-calibrated model may still err on high-cost examples.
>
> # Does NEC change model rankings?
>
> NEC vs error rate rankings are highly correlated (ρ=0.92–1.00), but NEC diverges meaningfully from other standard metrics: NEC vs balanced accuracy ρ=0.58 on Jigsaw and NEC vs Brier ρ=0.77–0.98 across datasets. NEC's primary value is not in reranking models relative to error rate, but in (1) revealing whether errors concentrate on low-cost examples and (2) providing different rankings from calibration and class-balance metrics that practitioners also commonly use.
>
> # Risks of treating high-agreement examples as high-cost
>
> While this risk is somewhat orthogonal to the cost-sensitive classification (annotator demographic biases are a risk for standard misclassification error), for completeness, we will include a few strategies for risk mitigation. First, we highlight the importance of recruiting a diverse annotator pool and assigning a diverse set of annotators to each data point. Second, keyword and semantic analyses can be used to test correlations between annotator demographic characteristics and annotator evaluations for content clusters.

---

> > ### Author Rebuttal · Reviewer_vMts · 2026-04-05
> >
> > The authors addressed my concerns. Therefore, I would retain my rating of weak accept.

---

### Decision · Program_Chairs · 2026-04-30

**Decision:**

Accept (regular)

**Comment:**

This paper introduces Normalized Excess Cost (NEC), a new evaluation metric for binary classifiers that weights misclassification errors by instance-level costs. These costs are derived from sources like annotator disagreement, distance to a decision threshold, or model confidence. The key empirical finding is that NEC is consistently lower than standard error rates, indicating that model errors cluster on ambiguous, low-cost examples. The paper further investigates whether these costs can be leveraged for training and finds inconsistent or negligible gains with standard cost-sensitive techniques, suggesting that cost-weighting is most effective when costs are predictable from input features.

The paper received four reviews: 1x Accept (5), 2x Weak Accept (4), and 1x Weak Reject (3). All reviewers acknowledged the clarity of the presentation and the practical importance of the problem. The reviewers exchanged their thoughts with the authors in the rebuttal process, consequently, all four reviewers marked their concerns as "Fully resolved".

I think that the reviewers' primary concerns regarding validation, novelty, and experimental thoroughness were directly and effectively addressed in a strong rebuttal. The paper's contributions, i.e., a new evaluation lens and a well-documented negative result for cost-sensitive training, are sufficiently significant for the ICML community.

Therefore, I vote for acceptance.